# GNeRV: A Global Embedding Neural Representation For Videos

## Abstract

In recent years, Implicit Neural Representation (INR) has garnered considerable attention for its effectiveness in compressing various visual information while delivering significant advantages in decoding speed. Video compression work with INR use time index as input and corresponding frame in RGB format as output. However, related work suffers from poor representation performance due to insufficient information in the embedding structure. In this paper, we introduce a global embedding structure, whose parameters are generated by random initialization and back propagation without any other constraint, and this embedding is shared by all frames. Furthermore, we propose a progressive training pipeline wherein large models are built upon the reuse and expansion of small models. Our **G**lobal embedding **Ne**ural **R**epresentation for **V**ideos (**GNeRV**) achieves SOTA results on multiple datasets. Taking UVG dataset as an example, GNeRV model outperforms the previously leading model HiNeRV by 1.5-2 dB at the same bitrate. And our progressive pipeline can effectively reduce the computational complexity of multi-bitrate encoding and save the storage space of multi-bitrate compressed files.

## 1 Introduction

The purpose of the video encoding is to compress the video in a way that ensures quality and facilitates its storage and transmission. Traditional video encoding methods, like H. 264 (Wiegand et al., 2003), HEVC (Sullivan et al., 2012), and VVC (Bross et al., 2021) typically rely on manually designed structures to compress video content across multiple dimensions. And with the development of deep learning, deep learning based approaches, such as DVC (Lu et al., 2019), DCVC (Li et al., 2021), and FVC (Hu et al., 2021) focus on replacing traditional components with deep learning modules. While these approaches have demonstrated the potential of neural network to achieve impressive rate-distortion performance, their codec have millions of parameters and decoding a pixel requires up to a million multiplications.

More recently, Implicit Neural Representations (INR) like NeRF (Mildenhall et al., 2021) have emerged as a captivating area of study for researchers. INR methods use neural networks to effectively represent diverse visual scenes by fitting them closely. This approach tends to have a very low complexity on the decoding side. As per the latest research in image INR Ladune et al. (2022), the complexity of decoding a pixel is only 0.7kMAC.

One prominent INR-based approach for video coding that has garnered considerable attention is NeRV (Chen et al., 2021). NeRV represents a video as a set of neural network weights. The input to the model is time index and the output is the corresponding whole image in RGB format. The network architecture combines Multi-Layer Perceptrons (MLP) and convolutional layers. Such lightweight architecture makes the decoding process computationally efficient and device-friendly.

However, the structure employed in current video INR work for embedding extraction presents certain challenges. NeRV (Chen et al., 2021) uses a huge MLP to extract frame-by-frame embedding from content-independent time vectors, and the whole process is inefficient. HNeRV (Chen et al., 2023) uses a convolutional encoder to extract frame-by-frame embedding from the image, but its embedding is not large enough, which makes the representation of the subsequent network more difficult. FFNeRV (Lee et al., 2022) learns the sparse frame-level embedding directly, and interpolate such sparse embedding to obtain the frame-by-frame embedding. Considering that the embedding

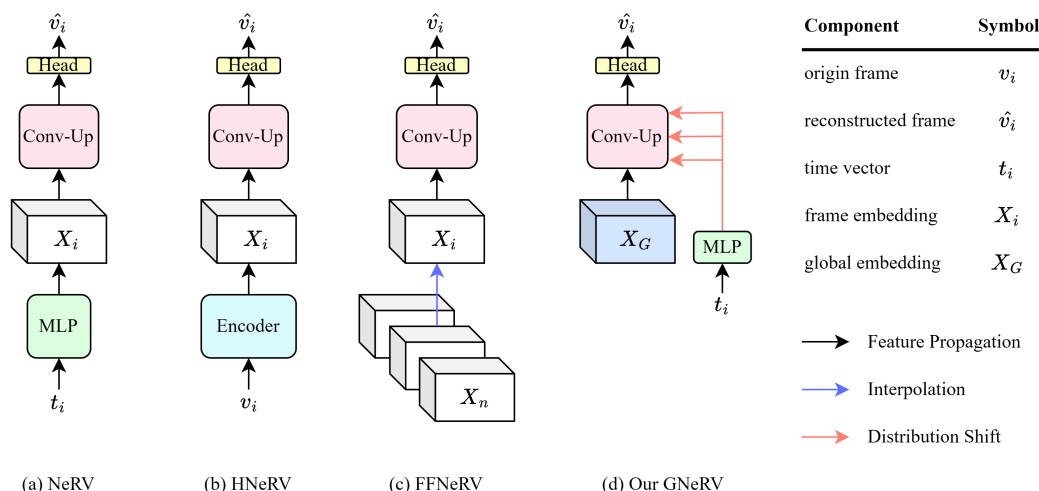

Figure 1: Comparison with several recent methods. From left to right: NeRV (Chen et al., 2021), HNeRV (Chen et al., 2023), FFNeRV (Lee et al., 2022) and our GNeRV.

of the video INR has little correlation with the final output, this method of introducing temporal information is generally ineffective.

What's more, the encoding side usually has to encode video at multiple bitrates to satisfy multiple decoding side requirements. However, in previous video INR work, models for each bitrate point had to be trained independently, and the models of high bitrate are not associated with the models of low bitrate, which leads to a waste of computational resources, as well as the storage space of the encoded files.

In this paper, we introduce a global embedding structure with parameters generated through random initialization and backpropagation, without any additional constraints, and it is shared by all frames. This structure can be directly input into the upsampling block and thus we don't have to use any embedding extraction networks. And unlike other work, we introduce temporal information only by shifting the distribution of intermediate features. Furthermore, for the first time, we explore the potential of video INR in representing residual information. We propose a progressive pipeline that employs INR to represent multi-level residual information, obtains better model performance, effectively reduces the computational complexity and storage cost of multi-bitrate compression. Our Global Embedding Neural Representation for Video (GNeRV) demonstrates superior performance when compared to other video INR models. For instance, on the UVG dataset, GNeRV outperforms HiNeRV (Kwan et al., 2023) by 1.5-2 dB while maintaining the same bpp.

In summary, our contributions can be summarized as follows:

- We introduce a novel global embedding structure shared by all frames in the video. Our GNeRV network achieves state-of-the-art compression results in video INR field across multiple datasets.
- We propose a progressive pipeline that uses video INR to represent residual information. This pipeline offers several advantages, including improved performance, increased computational efficiency, and reduced storage space requirements in scenarios requiring multi-bitrate coding.

## 2  RELATED WORK

**Embedding In Implicit Neural Representation.** Implicit neural representations (INR) leverage neural networks to represent diverse visual scene information, including images (Dupont et al., 2021; 2022), videos (Chen et al., 2021; 2023), and 3D scenes (Mildenhall et al., 2021; Barron et al., 2021). Typically, INR network implementations are denoted as $f_\theta(x) = y$, where $x$ represents the

embedding of the visual scene, $y$ contains RGB information, and $\theta$ signifies the network's weight information. Initially, researchers in these directions adopted positional information as an embedding $x$ to facilitate learning of high-frequency scene details, as suggested by Tancik et al. (Tancik et al., 2020). However, compared to content-independent time vectors, content-relevant embeddings as the starting point of the model's forward propagation will effectively reduce the learning difficulty of the network and thus improve the model's representation performance. Content-relevant embedding can be obtained in a variety of ways, using manual methods of coupling image information and location coding like Chen et al. (2022), extracting it from images using specific encoders like Chen et al. (2023); Zhao et al. (2023), or through random initialization and back propagation like Ladune et al. (2022); Müller et al. (2022); Lee et al. (2022). Considering video INR work, methods that contain more constraints like Chen et al. (2022; 2023) when embedding acquisitions perform much less well than methods with fewer constraints like Zhao et al. (2023); Lee et al. (2022). For embedding: less constraints yields better performance.

**INR For Video Compression.** NeRV (Chen et al., 2021) is a popular video INR work, due to the huge amount of information in the video and the existence of a lot of intra-frame and inter-frame redundancy, the efficiency of this image-wise model with convolution layers will be far more than that of the previous purely linear layer pixel-wise model like Tancik et al. (2020); Sitzmann et al. (2020). Among the work inspired by NeRV, Bai et al. (2022) applied patch-wise INR to represent segmented video data. Chen et al. (2023) uses an encoder to extract embedding information from the image, and Zhao et al. (2023) uses two encoders to extract embedding information from the image and inter-frame residuals and fuses them as the final embedding. He et al. (2023) uses an encoder-decoder architecture, where the encoder encodes the first and last two frames in a gop, and the decoder performs motion estimation and warping of features at all scales. Lee et al. (2022) obtains sparse, multilevel embedding information through random initialization and back propagation and obtains temporally correlated embeddings through the use of linear interpolation in the temporal dimension. This work also attempts to exploit inter-frame correlation by using the network output optical flows and weighted file warping outputs. Li et al. (2022) adjusts the position of the pixel-shuffle layer within the block and reduces the number of channels in the middle of the block compared to the original block structure. This design reduces the number of parameters in the block structure, enhancing expression efficiency without compromising performance. And another temporal branch was introduced using the distribution shift method in GAN (Huang & Belongie, 2017). Kwan et al. (2023) utilizes a new block structure that takes inputs as bilinear interpolations of the previous layer's block outputs, combined with spatio-temporal information embeddings, eliminating the need for feature size multiplication by a pixelshuffle layer.

## 3 METHOD

### 3.1 GNeRV: GLOBAL EMBEDDING NeRV

The overall architecture of our GNeRV model is illustrated in Figure.2. The input of the network is time index $t$, and the output is the the reconstructed image of frame $t$. The beginning of the network's forward propagation is global embedding, which is part of the network's trainable parameters. When passing through the block structure, the size of the embedding increases step by step and the number of channels is changed. Meanwhile, the time information will shift the distribution of features in the process. In the end, the header layer transforms the last layer of features into a 3-channel RGB format image.

**Embedding.** We discard embedding extraction module and instead introduce the embedding structure directly. The structure of embedding in our GNeRV model is a simple tensor, whose parameters will be randomly initialized and updated by back propagation. Its shape is $C_0 \times h \times w$, where $C_0$ is a hyperparameter that can be set in the configuration file. This parameter also directly determines the number of channels in the subsequent convolution block. The higher the number of channels, the better the model's representation. Here, $h, w$ is the shape of embedding, equal to the resolution of the video to be represented divided by its greatest common divisor. Taking the video whose resolution is $1080 \times 1920$ as an example, $h, w$ will be set to $9, 16$. Since all frames share one global embedding, this structure accounts for less than 1% of the total number of parameters in the model, saving the parametric quantities for later convolution networks.

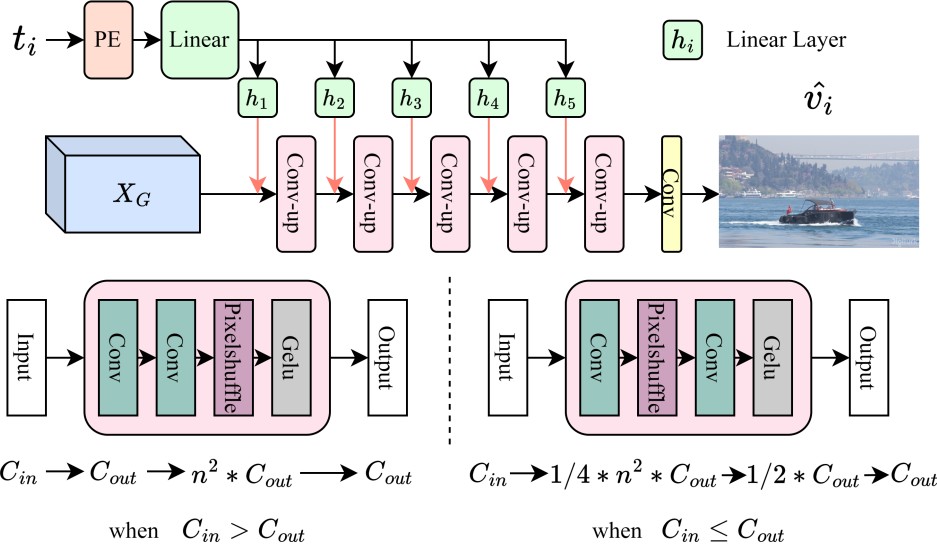

Figure 2: The detailed structure of GNeRV model, all frames share one global embedding. PE is short for position encoding. Red line represent the distribution shift operation. And there are two block structures under different conditions.

**Conv-Up Block.** The Conv-Up blocks are used for gradually multiplying feature sizes and restoring image information. By adjusting the upscaling factor in the configuration file, you can control the degree of feature size enlargement through the pixel shuffle layer. Conv-Up blocks are available in two structures, depending on the number of input and output channels. If the number of input channels is greater than the number of output channels, the block will be structured as Conv-Conv-Pixelshuffile, as depicted in the left block structure in Figure.2. Conversely, if the number of output channels is greater than the number of input channels, the block will be structured as Conv-Pixelshuffle-Conv, as shown in the right block structure.

**Distribution Shift Branch.** Because the upsampling process occurs incrementally, the output of each block becomes the input of the next block. Our time information is applied at the input of each block. The time index $t$ is passed through a positional encoding module and a small MLP layer that produces channel-wise means and variances, which are used to change the distribution of the input features.

$$DS\left(f_i^j\right) = \sigma_{i,j}\left(\frac{f_i^j - \mu\left(f_i^j\right)}{\sigma\left(f_i^j\right)}\right) + \mu_{i,j} \tag{1}$$

$$\mu_{i,j}, \sigma_{i,j} = h_j(h(\Gamma(t_i))) \tag{2}$$

$$\Gamma(t) = \left(\sin\left(b^0 \pi t\right), \cos\left(b^0 \pi t\right), \ldots, \sin\left(b^{l-1} \pi t\right), \cos\left(b^{l-1} \pi t\right)\right) \tag{3}$$

In Equation.1, $DS$ is short for distribution shift. $f_i^j$ represents the feature of the $i^{th}$ frame to be input into the $j^{th}$ upsampling block. $\mu(f_i^j)$ and $\sigma(f_i^j)$ denote the mean and variance of features $f_i^j$ across spatial dimensions, respectively. $\mu_i^j$ and $\sigma_i^j$ are generated by a small MLP as shown in Equation.2, with their input being the time index after position encoding as shown in Equation.3. $h$ represents the linear layer shared by all blocks, while $h_j$ represents the linear layer for the $j^{th}$ block. $b$ represents the base of position encoding, and $l$ represents the position encoding level. The length of the position encoded time vector is $2l$. We find that in video INR work, the efficiency of such distribution shifting method is much better than that of introducing temporal information from the embedding. More details will be shown in the ablation experiment.

**Model Compression.** At the end of training, we acquire model parameters in the form of 32-bit floating-point values. We perform parameter quantization tensor by tensor. Specifically, we calculate the channelwise maximum and minimum values of each tensor along the $0^{th}$ dimension and then uniformly quantize them using 8-bit quantization. After all the tensors have been processed, we will count the number of the quantized values and use arithmetic coding to entropy encode the quantized parameters to achieve further compression.

## 3.2 PROGRESSIVE PIPELINE

If the high bitrate model can utilize the already trained low bitrate model, the high bitrate model does not have to be trained from scratch. The encoding side will save considerable computing power and storage space, or achieve better compression performance with the same computational resources. We thus propose a progressive pipeline. Such pipeline is depicted in Figure 3. To illustrate this process, let's consider the three-stage pipeline as an example. In this approach, we utilize the first network (low bitrate) to represent the original video. At the end of training, we compute frame-by-frame differences between the original video and the reconstructed video as a residual video. This residual video then serves as the target for training the second network. Once the training of the second network (mid bitrate) is completed, we calculate the difference again. This time, it's between the real residual and the residual generated by the second network. We then initiate training for the third network (high bitrate). During the output phase, the process is reversed. The reconstructed video with low bitrate is combined with the multilevel residual video for the final output. Since the multilevel network can be decoded in parallel, there is no increase in decoding time, and such process is iterable as shown in Equation.4 and 5

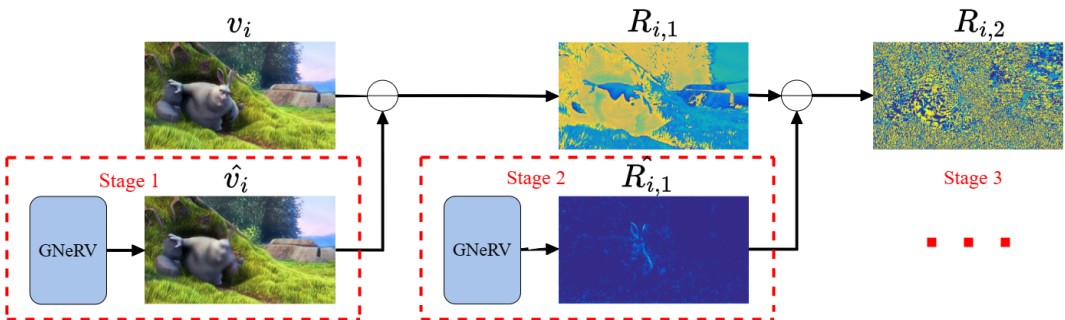

Figure 3: Pipeline of the 3-stage progressive process.

$$R_{i,j} = \begin{cases} R_{i,j-1} - \hat{R}_{i,j-1} & \text{if } j \geq 2 \\ v_i - \hat{v}_i & \text{if } j = 1 \end{cases} \tag{4}$$

$$\hat{Y}_{i,j} = \begin{cases} \hat{v}_i + \sum_{j=1}^{n} \hat{R}_{i,j} & \text{if } j \geq 1 \\ \hat{v}_i & \text{if } j = 0 \end{cases} \tag{5}$$

where $\hat{R}_{i,j}$ represents the $j^{th}$ level residual for the $i^{th}$ frame to be represented by INR. $\hat{Y}_{i,j}$ describes the final reconstruction of the $j^{th}$ level network for frame $i$. Variables with caps denote data reconstructed by INR, while those without caps represent real data.

## 4 EXPERIMENT

### 4.1 DATASETS AND IMPLEMENTATION DETAILS

We evaluated GNeRV on the Bunny sequence, UVG dataset (Mercat et al., 2020), and MCL-JCV dataset (Wang et al., 2016). The Bunny sequence comprises 132 frames with a resolution of 1280 x 720. UVG includes 7 videos with resolutions of 1920 x 1080, each having a length of 600 or

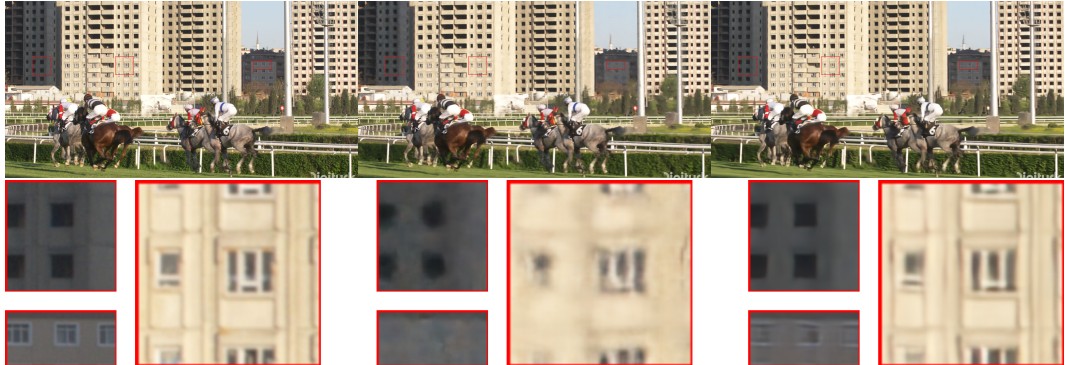

Figure 4: Frame 539 of the Ready. sequence from the UVG dataset. From left to right: original image, reconstructed image of HNeRV, reconstructed image of our GNeRV.

300 frames. The MCL-JCV dataset consists of 30 videos with a resolution of 1920 x 1080, each containing 150 frames.

During model training, we used the Adam optimizer with beta as (0.5, 0.999). Learning rate is set to 2e-3 for the original video and 2e-4 for the residual video in progressive pipeline, with warmup cosine learning rate decay, 20% warmup epochs. We trained the model for 300 epochs with a batch size of 4, unless specified otherwise. The embedding shape is $C_0 \times 9 \times 16$ since the video resolution ratio is 9:16, where $C_0$ is a hyperparameter defining the channel number for embedding and subsequent Conv-Up blocks, directly controlling the model's parameter count. The first block doubles the feature channel, and each subsequent block halves the channel count. When features' channels number decreases, the minimum channel number of the the block's output feature is maintained at 30. The upscaling factors of Conv-Up blocks for 1080p video are set to 5, 3, 2, 2, 2, and for 720p video, they are set to 5, 2, 2, 2, 2. For the position encoding in the distribution shift branch, $b$ and $l$ are set to 1.25 and 20, respectively. The input and output channels of common linear layer used for feature distribution shift are set to 40 and 40. And The input and output channels of the separate linear layer for each block are set to 40 and the channel of the input into each block.

The loss function of the model is defined in Equation 6:

$$L = \frac{1}{T} \sum_{i=1}^{T} \alpha \, \|\mathbf{v}_i - \hat{\mathbf{v}}_i\|_1 + (1 - \alpha)\,(1 - \text{SSIM}\,(\mathbf{v}_i, \hat{\mathbf{v}}_i)) \tag{6}$$

In Equation 6, $\mathbf{v}_i$ represents the ground-truth frame in the video, while $\hat{\mathbf{v}}_i$ represents the reconstructed frame. The term $\alpha$ serves as a trade-off factor between MSE loss and SSIM loss. In our experiments, $\alpha$ is set to 0.7 to balance the losses when processing full frames. When working with residual video, $\alpha$ is set to 1, as SSIM is irrelevant for residual information.

All experiments were conducted on a single Nvidia 4090 GPU using Pytorch version 2.1.0.

## 4.2 MAIN RESULT

**Video Compression.** We conducted a comprehensive comparison of our model with existing video INR models and traditional codecs on multiple datasets. Specifically, these models include NeRV (Chen et al., 2021), HNeRV (Chen et al., 2023), ENeRV (Li et al., 2022), DNeRV (Zhao et al., 2023), HiNeRV (Kwan et al., 2023), and traditional codecs H.264 (Wiegand et al., 2003) and H.265 (Sullivan et al., 2012).

Our model achieved state-of-the-art results for video compression in various scenes and at different resolutions, surpassing other video INR methods. Our network demonstrated adaptability to scenes with different resolutions.

On the Bunny sequence, we conducted a comparison of our GNeRV model with NeRV, HNeRV, and ENeRV. GNeRV outperformed all other models across all sizes, as depicted in Table.1. It's worth

noting that the parameter number of the E-NeRV model cannot be set below 2 million due to the complexity of its attention structure in the embedding generation module. And the quality of video reconstructed by NeRV model is poor at low bitrate. So the lower band of model parameter number is set to 0.57.

Table 1: Video representation results on the bunny sequence. Results are in PSNR.

| | | | | | | | | | | |
|---|---|---|---|---|---|---|---|---|---|---|
| NeRV | Parameter Number/M | - | - | - | 0.57 | 0.95 | 1.51 | 3.2 | 6.8 | 12.52 |
| | PSNR(dB) | - | - | - | 27.71 | 29.11 | 30.85 | 33.85 | 36.81 | 39.56 |
| HNeRV | Parameter Number/M | 0.058 | 0.124 | 0.291 | 0.545 | 0.96 | 1.5 | 3.575 | 6.77 | 12.5 |
| | PSNR(dB) | 27.27 | 27.31 | 29.76 | 31.55 | 33.65 | 35.58 | 38.51 | 40.63 | 42.31 |
| E-NeRV | Parameter Number/M | - | - | - | 2.25 | 2.53 | 2.86 | 3.5 | 6.81 | 12.5 |
| | PSNR(dB) | - | - | - | 31.19 | 33.93 | 35.52 | 37.19 | 40.62 | 42.86 |
| GNeRV | Parameter Number/M | **0.053** | **0.115** | **0.274** | 0.519 | 0.92 | 1.66 | 3.17 | 6.67 | 12.42 |
| | PSNR(dB) | **29.11** | **31.95** | **34.19** | **35.73** | **37.46** | **39.01** | **40.83** | **42.58** | **43.53** |

On the UVG dataset, we compared our GNeRV model with NeRV, HNeRV, DNeRV, HiNeRV and x264, x265. Note that HNeRV and DNeRV can only reconstruct videos with a resolution ratio of 1:2 due to their fixed embedding generator, leading to representing on center-cropped videos. As for the traditional codec, we utilized the ffmpeg library with the medium preset and no additional restrictions. GNeRV consistently outperformed all other INR methods across all sizes, and GNeRV model exhibited superior rate-distortion (R-D) performance compared to traditional codecs at low bitrate. The result is shown in Figure.5. We replicated the NeRV, HNeRV experiments, and Table.2 demonstrates the sequence-level quality comparisons of the four size models. Note that when the model parameter number becomes large, the training of HNeRV becomes unstable. And we replaced these unstable results using the previous best recovery quality, indicated using italics.

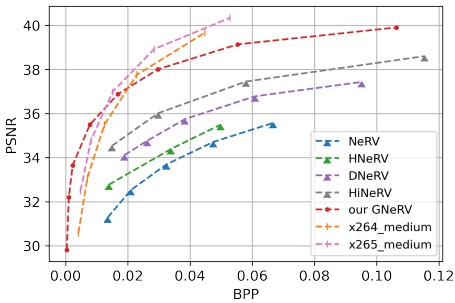 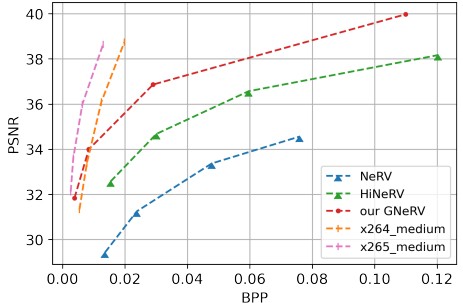

Figure 5: The rate-distortion curves on UVG dataset

Figure 6: The rate-distortion curves on MCL-JCV dataset

Table 2: Video representation results on the UVG dataset. MAC is short for Multiplication and Accumulation. Results are in PSNR(dB).

| Model | Size | MACs/G | FPS | Beauty | Bosph. | Honey. | Jockey | Ready. | Shake. | Yacht. |
|---|---|---|---|---|---|---|---|---|---|---|
| NeRV | 0.202M | 7.30 | 320 | 29.45 | 27.93 | 31.14 | 24.49 | 19.06 | 27.99 | 24.11 |
| HNeRV | 0.210M | 25.16 | 270 | 29.82 | 28.29 | 33.5 | 24.11 | 18.97 | 29.71 | 24.31 |
| GNeRV | **0.058M** | 10.59 | 140 | **32.73** | **31.89** | **36.18** | **26.12** | **22.67** | **31.26** | **27.8** |
| NeRV | 4.61M | 13.90 | 295 | 33.66 | 34.39 | 38.42 | 33.06 | 26.22 | 34.32 | 29.33 |
| HNeRV | 4.47M | 329.70 | 54 | 33.89 | 35.45 | 39.4 | 32.87 | 26.56 | 35.44 | 30.08 |
| GNeRV | 4.66M | 54.80 | 141 | **40.13** | **40.76** | **42.96** | **37.06** | **30.91** | **39.84** | **34.39** |
| NeRV | 8.63M | 25.21 | 250 | 34.03 | 35.9 | 39.09 | 34.9 | 28.22 | 35.62 | 30.99 |
| HNeRV | 8.66M | 640.10 | 35 | 34.21 | 36.86 | 39.62 | 34.71 | 28.64 | 36.52 | *30.08* |
| GNeRV | 8.72M | 85.55 | 140 | **40.42** | **41.77** | **43.07** | **39.35** | **33.13** | **40.5** | **35.71** |
| NeRV | 14.02M | 45.01 | 190 | 34.27 | 37.17 | 39.44 | 36.14 | 29.93 | 36.69 | 32.35 |
| HNeRV | 14.03M | 1041.04 | 22 | 34.37 | *36.86* | 39.35 | 35.91 | 30.23 | 37.37 | *30.08* |
| GNeRV | 14.08M | 133.07 | 143 | **40.57** | **42.41** | **43.19** | **40.52** | **34.79** | **41.26** | **36.58** |

Table 3: Comparison of computational complexity, storage space, reconstruction performance betweeen two pipelines.

| Pipeline style | | Stage 1 | Stage 2 | Stage 3 | Total |
|---|---|---|---|---|---|
| Respective | Channel Number | 160 | 230 | 280 | - |
| | MACs/G ↓ | 54.80 | 91.14 | 133.07 | 279.01 |
| | Parameter Number/M ↓ | 4.66 | 9.49 | 14.07 | 28.22 |
| | PSNR(dB) ↑ | 38.01 | 38.91 | 39.90 | - |
| Progressive | Channel Number | 160 | 160 | 160 | - |
| | MACs/G ↓ | 54.80 | 54.80 | 54.80 | **164.4** |
| | Parameter Number/M ↓ | 4.66 | 4.66 | 4.66 | **13.98** |
| | PSNR(dB) ↑ | 38.01 | **39.60** | **40.42** | - |

On the MCL-JCV dataset, we compared our GNeRV model with NeRV, HiNeRV and traditional codec. Our model consistently outperforms other video INR models and is comparable to conventional encoders at low bitrate as shown in Figure.6. Comparing the performance of the model on both datasets, the video INR work is more advantageous when dealing with longer sequences.

**Progressive Pipeline.**

We assumed that the coding side needs to produce compressed files with low, medium and high bitrate. We used two processes to realize this need, Method 1 is to obtain implicit neural representations of the three qualities using small, medium and large models trained from the scratch respectively, referred to as "Respective". Method 2 adopts progressive process, where the model's training target is the residual information of the original video and the reconstructed video from the previous models, referred to as "Progressive".

Such experiments were conducted on the UVG dataset. We used the three-stage progressive pipeline on the base GNeRV model with 160 channels, whose corresponding bpp is around 0.03. Based on the model we had already gained, we trained two other models of the same size for representing the two-level residuals, with 300 epochs of training for each model. We can reuse the model at low bitrate, and finally get a better reconstruction result with a model trained from scratch at the same bitrate. What's more, progressive pipeline can reduce computational complexity and save storage space comparing to the pipeline that train every model in a single stage. We show the differences between the two approaches in the Table.4.2.

### 4.3 ABLATION STUDY

**Training Epochs.** Given the nature of INR work, which focuses on overfitting, the number of training epochs directly affects the final model performance. We performed controlled experiments where we varied the number of training epochs on the Bunny sequence. The results are illustrated in Figure.7.

**Temporal Information Coupling Path.** The method that incorporates time information into the neural network through the distribution shift of feature is referred to as "DS". And we designed an ablation experiment to argue that such design is much more efficient than modulating the embedding with time information directly, referred to as "M". Specifically, after reshaping the time vector to $(1, 9, 16)$, we transformed it into temporal information with the same number of channels as the embedding through two convolutional layers, and added this tensor to the embedding to complete the modulation. We control the parameter number of DS-branch and that of M-branch the same.

We conducted this experiment on the Bunny sequence and obtained the results shown in Table.4. Our original model contains only DS-branch. We trained a model with two time-informative branches. We used this model to produce three outputs, a direct output, an output that masks the M-branch, and an output that masks the DS-branch. When comparing the model with two branches, it was observed that the DS-branch had a significant influence, while masking the M-branch had minimal impact on the output. When comparing the original model and the model with two branching paths, the two temporal branching paths introduced more parameters while making the model degrade at high bitrate, which suggests that modulating the embedding directly instead makes training difficult.

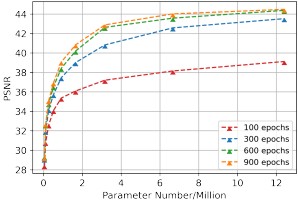 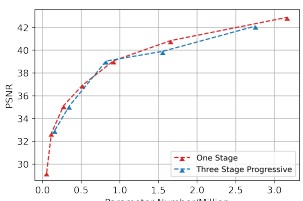 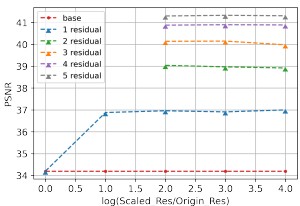

Figure 7: The effect of the number of training epoch on the video quality reconstructed by the model.

Figure 8: Comparison between progressive pipeline and one-stage pipeline.

Figure 9: A 40-channel model representing bunny sequence with different scale in progressive pipeline.

Table 4: Ablation study on the time branch conducted on bunny sequence. Results are in PSNR(dB).

| Channel Number | 20 | 40 | 80 | 152 | 223 | 304 |
|---|---|---|---|---|---|---|
| Parameter Number/M | 0.115 | 0.274 | 0.92 | 3.17 | 6.67 | 12.42 |
| w. DS | 31.95 | 34.19 | **37.46** | 40.83 | **42.58** | **43.53** |
| w. DS and M | **32.00** | **34.26** | **37.46** | **41.12** | 42.46 | 43.41 |
| w. DS and M, mask M | 31.98 | 34.22 | 37.42 | 40.86 | 42.14 | 42.26 |
| w. DS and M, mask DS | 17.93 | 17.96 | 17.96 | 17.88 | 17.95 | 17.97 |

**Progressive Pipeline.** When we adopt the progressive pipeline to train our INRs at one bitrate, it consumed more training epochs to represent the same video sequence. Thus, we need to conduct experiments to ensure that the pipeline remains efficient compared to the original model. Specifically, since we trained a base model for 300 epochs and two residual models for 600 epochs in a 3-stage progressive pipeline, we trained a base model for 900 epochs for comparison on the Bunny sequence. The results are presented in Figure.8. We found that such a pipeline does not lead to performance degradation. This implies that we can confidently use it to gain engineering convenience.

**Scale Of Residual Information.** When employing a network to represent residual information, it's beneficial to use a small trick: scaling the residuals before training. This ensures that INR can effectively learn valid residuals. Figure.9 illustrates how the scale of residuals impacts the results. The horizontal axis represents the logarithm of the scale of the residuals. We experimented with scales of 1, 10, 100, 10k, and 100k on the first level of residuals. The second and subsequent levels of residuals are based on the results calculated by multiplying the previous level of residuals. The residual scale remains constant at 100 and does not multiply level by level; instead, it represents an absolute. We found that only the first residual requires a scaling factor to handle the residual information effectively. There is no need to use larger residual scales for subsequent INR levels to avoid performance degradation. Therefore, we set the default scale for residuals to 100.

## 5 CONCLUSION

In this paper, we introduce GNeRV, a novel video implicit representation model that builds upon the input-output format of the NeRV framework while significantly enhancing the embedding structure. Our main innovation lies in the incorporation of a global embedding structure, which replaces the entire architecture for embedding extraction. GNeRV boasts a straightforward design that adapts seamlessly to video sequences of varying resolutions, and it allows for model size to be scaled efficiently from nearly zero, showcasing remarkable potential for low-bitrate applications. Our approach attains state-of-the-art results across multiple datasets. Additionally, we present a novel training pipeline in which we utilize INR to overfit residual information, offering support for multiple-bitrate coding scenario, where high bitrate models can reuse low bitrate models. We can obtain better representation performance and reduce the computational complexity. Lastly, we emphasize that our method serves as an effective alternative to traditional techniques in offline compression scenarios for low-bitrate videos.

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
