# OpenReview forum: "GNeRV: A Global Embedding Neural Representation For Videos"
_ICLR.cc/2024/Conference — ICLR 2024 Conference Withdrawn Submission_

### Official Review · Reviewer_ggkq · 2023-10-16

**Soundness:** 3 good
**Presentation:** 2 fair
**Contribution:** 3 good
**Rating:** 6
**Confidence:** 4

**Summary:**

1. This paper investigates the problem of learnt video compression based on implicit neural representations (INR).
2. A global embedding is introduced, which is shared by all frames and optimized through backpropagation.
3. This global embedding is fed to the decoder together with the frame index, without the need of encoder.
4. The proposed GNeRV outperforms HiNeRV and achieves SOTA RD performance over multiple datasets like UVG and MCL-JCV.
5. A progressive residual coding pipeline is incorporated, leading to more efficient multi-bitrate encoding and better multi-bitrate RD performance.

**Strengths:**

strengths:

Overall, this is a solid paper.
The method and experiment are all clearly explained.
Most previous works are compared properly with multiple datasets.
To my knowledge, the global embedding proposed in this paper is novel in this INR for video compression field.
The RD performance is better than previous SOTA.

**Weaknesses:**

weakness:

The contribution part should be improved to better explain the main contributions of this paper such as the significance of the RD improvement and multi-bitrate compression. Currently there are only two bullets.

Introducing and optimizing the global latent is quite related to the explicit bit allocation framework proposed in Xu, T., Gao, H., Gao, C., Wang, Y., He, D., Pi, J., ... & Zhang, Y. Q. (2023, July). Bit allocation using optimization. In International Conference on Machine Learning (pp. 38377-38399). PMLR. In my understanding, GNeRV can be considered as implicit bit allocation using optimization. Please discuss this in the revised version to help readers build deeper understandings regarding this field.

More visualizations should be provided to compare with previous works.

**Questions:**

Please refer to the weakness part. And:

In Table 2, why the complexity of DNeRV and HiNeRV are not reported? Also, the FPS of x264 abd x265 should be provided for better comparison.

In Table 1, why HiNeRV, x264, x265 are not reported?

---

### Official Review · Reviewer_nVh1 · 2023-10-30

**Soundness:** 3 good
**Presentation:** 3 good
**Contribution:** 3 good
**Rating:** 6
**Confidence:** 4

**Summary:**

The paper proposes a progressive approach to implicit neural representation compression for videos. It proposes a global embedding, obtained by an MLP, for representing the time/frame index joint with a global representation for the video frame content. Learning the progressive representation is obtained by training the GNeRV for residuals of multiple stages. The results are interestingly good for high-bitrate case in comparison to other INR-based approaches.

**Strengths:**

- The paper is fairly well-written and easy to follow.
- Training a INR that is handling multiple rates using a single representation is interesting and novel angle in this paper.
- The results are conclusive and supporting the hypothesis.
- The ablation studies are sufficient to demonstrate the contribution of various components and steps of the method.

**Weaknesses:**

- While conventional video codecs are tested, the results for other neural approaches are missing. That could improve the paper in terms of conducted experiments.

**Questions:**

- Why for different size of models and increased MACs the FPS of proposed method is constant and even improves? This could be a nice addition in discussions about the proposed method.
- Figure 3 could be improved by having training and inference pipeline better clarified.
- Is there a particular reason for not reporting the other neural based video coding approaches?

---

### Official Review · Reviewer_vfx3 · 2023-10-31

**Soundness:** 3 good
**Presentation:** 3 good
**Contribution:** 2 fair
**Rating:** 3
**Confidence:** 3

**Summary:**

The authors propose a novel implicit neural representation model for video compression. Unlike HNeRV (CVPR 2023), the proposed method doesn't require an encoder module for embedding extraction for each frame. Instead, it directly optimizes one global embedding for an entire video and adapts it for different frames using the proposed distribution shift branch with temporal index as input. Additionally, the authors introduce a progressive training strategy, enabling multiple rates to be achieved based on multiple stages of residual information fitting. Experimental results on Bunny, UVG dataset, and MCL-JCV dataset demonstrate superior performance compared to other methods.

**Strengths:**

- The paper is well-written and easy to follow.
- The study features extensive experiments and achieves superior results.

**Weaknesses:**

- How are the variable compression rates controlled using the proposed progressive pipeline? We know it is feasible to achieve this using the 'respective' approach. Further clarification would be beneficial.
- The objective function combines MSE and SSIM, however, all quantitative results are reported in terms of PSNR. Typically, for optimal performance on each metric, it is advisable to use only the corresponding one. Please clarify its motivation, and include the comparison in terms of SSIM.
- While HiNeRV outperforms h265, it performs significantly worse compared to h265 in this context. The authors should clarify the details of the experimental setting.

**Questions:**

My main concern pertains to the novelty of the proposed approach. The distribution shift branch appears to employ a technique same as to COIN++. Although the proposed method targets video compression while COIN++ focuses on dataset compression, the reliance on the distribution shift is crucial for me. I think similar performance could be achieved by employing a grid input (as in HiNeRV) with distribution shift, instead of using a global embedding, which is the main contribution emphasized by the authors.

---

### Official Review · Reviewer_qKqs · 2023-11-01

**Soundness:** 2 fair
**Presentation:** 1 poor
**Contribution:** 2 fair
**Rating:** 3
**Confidence:** 3

**Summary:**

The authors introduce a new video representation and compression technique built upon Implicit Neural Representation (INR). Along with giving time index as input and RGB format as output, the authors propose the GNeRV technique which gives a global representation shared amongst all frames. A tensor based embedding structure is introduced whose shape is determined by a hyperparameter and video’s resolution. Time information is used at each block’s input and positionally encoded data is applied to measure shift. A progressive pipeline is used to eventually form a high-bitrate model. In detailed ablation study is conducted to evaluated each components significance. Extensive experiments have been conducted to compare with other state of the art video compression models.

**Strengths:**

1. Novel compression techniques have been applied - progressive pipeline and quantization
2. In-depth ablation studies have been conducted to compare various aspects of the model. Extensive experimentation has been conducted on several aspects of their own pipeline and compared with other state-of-the-art models.

**Weaknesses:**

1. “The input of the network is time index t, and the output is the reconstructed image of frame t” - Why is t used as a parameter for both?
2. Is there a variance observed in the PSNR values across different runs?
3. Residual information is defined before using it in the introduction
4. Figure -  X_G in the figure is not defined. The description of the figure could be better. There are symbols that are not self-explanatory and need to refer to various different sections to get a better understanding of the model architecture. It would be great if we can mark several components of the architecture as named in the sections in the figure.
5. Writing improvements - There are several terms whose definition is missing or have been used loosely like header layer.

**Questions:**

1. How much overhead does residual computation cause, especially for long-form videos?
2. " Since all frames share one global embedding, this structure accounts for less than 1% of the total number of parameters in the model,
saving the parametric quantities for later convolution networks", is this true for all the different hyperparameter settings used in the experiments?